# Independent Membrane Binding Properties of the Caspase Generated Fragments of the Beaded Filament Structural Protein 1 (BFSP1) Involves an Amphipathic Helix

**DOI:** 10.3390/cells12121580

**Published:** 2023-06-07

**Authors:** Miguel Jarrin, Alexia A. Kalligeraki, Alice Uwineza, Chris S. Cawood, Adrian P. Brown, Edward N. Ward, Khoa Le, Stefanie Freitag-Pohl, Ehmke Pohl, Bence Kiss, Antal Tapodi, Roy A. Quinlan

**Affiliations:** 1Department of Biosciences, Upper Mountjoy Science Site, The University of Durham, South Road, Durham DH1 3LE, UKr.a.quinlan@durham.ac.uk (R.A.Q.); 2Biophysical Sciences Institute, Durham University, Upper Mountjoy, South Road, Durham DH1 3LE, UK; 3Department of Biological Structure, University of Washington, Seattle, WA 98195, USA; 4Department of Chemistry, Durham University, Lower Mountjoy, South Road, Durham DH1 3LE, UK; 5Department of Biochemistry and Medical Chemistry, Medical School, University of Pécs, 7624 Pécs, Hungary

**Keywords:** lens, cataract, intermediate filaments, amphipathic helix, oligomerisation, intrinsically disordered domain, lipid binding, post-translational myristoylation, membrane scaffold

## Abstract

Background: BFSP1 (beaded filament structural protein 1) is a plasma membrane, Aquaporin 0 (AQP0/MIP)-associated intermediate filament protein expressed in the eye lens. BFSP1 is myristoylated, a post-translation modification that requires caspase cleavage at D433. Bioinformatic analyses suggested that the sequences 434–452 were α-helical and amphipathic. Methods and Results: By CD spectroscopy, we show that the addition of trifluoroethanol induced a switch from an intrinsically disordered to a more α-helical conformation for the residues 434–467. Recombinantly produced BFSP1 fragments containing this amphipathic helix bind to lens lipid bilayers as determined by surface plasmon resonance (SPR). Lastly, we demonstrate by transient transfection of non-lens MCF7 cells that these same BFSP1 C-terminal sequences localise to plasma membranes and to cytoplasmic vesicles. These can be co-labelled with the vital dye, lysotracker, but other cell compartments, such as the nuclear and mitochondrial membranes, were negative. The N-terminal myristoylation of the amphipathic helix appeared not to change either the lipid affinity or membrane localisation of the BFSP1 polypeptides or fragments we assessed by SPR and transient transfection, but it did appear to enhance its helical content. Conclusions: These data support the conclusion that C-terminal sequences of human BFSP1 distal to the caspase site at G433 have independent membrane binding properties via an adjacent amphipathic helix.

## 1. Introduction

The transparency and optical function of the eye lens depends upon a functional intermediate filament network [1,2]. In addition to vimentin, there is a pair of intermediate filament proteins that are highly expressed in the lens and form an iconic cytoskeletal structure of the eye lens called the beaded filament [3,4]. For this reason, they are identified as Bead Filament Structural Proteins 1 and 2 [5]. Mutations in both proteins cause cataracts ((https://cat-map.wustl.edu/, accessed on 5 May 2023); [6]), and the targeted removal from the mouse genome not only affects the biomechanical properties of the lens [7], but also affects the transparency and optical properties of the lens ([8,9,10,11,12]; reviewed in [1,2,13,14]) and their loss can also prevent emmetropia developing [15]. BFSP1 (beaded filament structural protein 1; filensin; CP95) and its orthologues are found from cephalopods to mammals [16]. Similar to other intermediate filament proteins, BFSP1 has a central α-helical rod domain flanked by N- and C-terminal domains, although the C-terminal domain is striking because it exhibits lower sequence conservation between species ([2,17]; Appendix A), which is not seen for other intermediate filament proteins such as vimentin. There are, however, some sequences that are highly conserved across species [18,19] and these include the site of a caspase cleavage site (DVPD; human BFSP1 430–433) and adjacent myristoylation signal ([19,20]; (GGQISK; human BFSP1 434–439)). Post-translational myristoylation following caspase cleavage is a recognised mechanism for adding this fatty acid to an exposed N-terminal glycine [21,22] so much so that it is incorporated into computational tools to predict myristoylation sites in the human reference proteome with BFSP1 (Q06002) identified as a probable hit [23]. Imaging mass spectrometry has confirmed BFSP1 myristoylation occurs at G434 following caspase cleavage in BFSP1 [18,24]. This post-translational myristoylation event occurs in a distinct region of the lens as the transition from vimentin to beaded filaments completes [25,26]. This represents an important transition point for the lens, but also because it is the first myristoylation event to be identified in the spectrum of intermediate filament post-translational modifications [27,28,29,30].

The cleavage, release and myristoylation of BFSP1 at G434 then contribute to the regulation of the major water channel protein, AQP0, in the lens [18]. AQP0 and BFSP1 interact directly [31], as shown by chemical cross-linking studies that identified the interaction to occur between BFSP1 450–465 and AQP0 239–259 [31]. The site was identified after tryptic digestion of the cross-linked sample. This demonstrates that BFSP1 binds the C-terminal region of AQP0, a domain that is crucial to its water channel regulation [32,33]. This observation implies that AQP0 could facilitate the binding of BFSP1 [20,24,26,34,35] and its myristoylated fragments [18] to the lens fibre cell plasma membranes. These BFSP1 fragments derived from the C-terminus are tightly associated with the lens plasma membranes and resist extraction with both urea and sodium hydroxide [18], a biochemical property of integral membrane proteins [36]. Myristoylation is an important regulatory post-translation modification that helps target proteins to specific membranes and subcellular compartments [21,37,38]. In the case of some membrane-targeted proteins, such as Adenosine diphosphate Ribosylation Factors (ARF), myristoylation can, however, affect the conformation of the adjacent peptide and change its affinity for the membrane [39,40] and can determine its membrane association [41]. It is important, therefore, to determine whether the caspase released and myristoylated fragments of BFSP1 do or do not require AQP0 for their avid association with lens membranes or whether this property is attributable to sequences/motifs in the C-terminal sequences of BFSP1 itself.

Although there is no crystallographic data for BFSP1, an AlphaFold2 prediction is available (https://alphafold.ebi.ac.uk/entry/Q12934, accessed on 5 May 2023). AlphaFold2 predicts that the BFSP1 tail domain is mostly intrinsically disordered, but it does predict a short α-helix at the cryptic myristoylation site, G434. Structural data are available for ARF proteins [40,41], and in their case, an amphipathic helix is preceded by N-terminal glycine that is myristoylated. An amphipathic helix behaves similarly to an integral membrane protein in terms of its urea/alkali resistance to extraction from membranes [42], suggesting a possible mechanism to explain how the BFSP1 C-terminal domain fragments associate with the lens membranes. It remains to be determined whether the AlphaFold2 prediction for a short α-helix and whether an amphipathic helix could be formed by the residues immediately distal to the myristoylation site G434 in human BFSP1.

Here we investigate the membrane binding properties of BFSP1 and C-terminal domain fragments generated by caspase cleavage. Using chemically synthesised polypeptides, we show that the BFSP1 region 434–467 adopts an α-helical secondary structure in the presence of trifluoroethanol (TFE) that is enhanced by the presence of a myristoyl group. Using recombinantly expressed and purified fragments from the C-terminal domain of BFSP1, we use surface plasmon resonance (SPR) to measure a sub-micromolar K_D_ for the BFSP1 fragment 434–548. Transient transfection of tissue culture cells with eGFP-tagged constructs demonstrates that providing the 434–460 sequence was present, the transfected C-terminal BFSP1 fragment from the C-terminal domain would localise to the plasma membranes and some intracellular membrane compartments. Our data support the conclusion that myristoylation occurs at the N-terminus of an amphipathic helix, and any BFSP1 C-terminal domain fragments containing this region have the capacity to associate with cell membranes independent of the lens water channel protein AQP0.

## 2. Materials and Methods

### 2.1. Bioinformatic Analyses

Potential lipid binding regions in the C-terminal region of BFSP1 were identified using HELIQUEST ([43]; https://heliquest.ipmc.cnrs.fr, accessed on 10 March 2023; version 2.0, Nice, France). The analysis assumed an α-helix and analysed the sequence using an 18-residue window. The “Discrimination factor” was calculated as described [44] using the formula D = 0.944 (<µH>) + 0.33 (*z*), where µH is the calculated hydrophobic moment and *z* is the calculated net charge. The combined use of the HELIQUEST discrimination factor and the Eisenberg plot methodology is a robust bioinformatics approach to identifying lipid-binding regions [45]. The fIDPnn webserver (http://biomine.cs.vcu.edu/servers/flDPnn/, accessed on 17 February 2023; version 1, Richmond, VA, USA) was used to assess the intrinsic disorder of the same region [46]. The MemBrain 3.0 webserver was used to assess the amphipathic helix-forming propensity of the C-terminal domain of BFSP1 (http://www.csbio.sjtu.edu.cn/bioinf/MemBrain/, accessed on 17 February 2023; version 1; Shanghai, China [47]). Sequence alignment of human (Q12934 BFSP1_Human), bovine (Q06002 BFSP1_BOVIN) and rat (Q02435 BFSP1_Rat) downloaded from UniProt [48] was produced using COBALT [49]. AlphaFold2, a protein structure predicting algorithm, was used to predict secondary sequence features of the BFSP1 C-terminal sequence 434–665 [50,51]. The predicted structure was viewed, and images of this region were produced using iCn3D [52].

### 2.2. Cloning, Expression and Purification of Human BFSP1, Its Fragments and CRYAB

These experimental details have been given in detail previously [18]. Briefly, full-length human BFSP1 was originally obtained from Source Bioscience (http://www.lifesciences.sourcebioscience.com/IMAGE, accessed on 17 February 2023, clones 6154051 and 5406467; Nottingham, UK). Constructs encompassing and within the domain 434–665 of BFSP1 were generated by PCR and, after sequencing, were sub-cloned into pET23b vectors to include a C-terminal hexa-histidine tag and the residues LE encoded by the XhoI site where the amplified fragments were cloned in frame with the His-tag of the vector. Recombinant His-tagged BFSP1 fragments were purified by affinity chromatography on nickel columns according to the manufacturer’s instructions (His-Select Nickel Affinity Gel; Sigma-Aldrich, Poole, UK) and with a final purification step using a size exclusion column (Sephadex G10; Sigma-Aldrich, Poole, UK). In some instances, a Superose 6 10/300 column (Cytiva Lifesciences, Cardiff, UK) was used as a final purification step. Fractions containing the protein of interest were pooled and concentrated in an Amicon Stirred Cell and ultracell regenerated cellulose membranes with 10 and 3 kDa cut-off as appropriate. Recombinant human CRYAB was also produced recombinantly in *E.coli* as described previously [53]. Native bovine BFSP1 was purified from eye lenses as described previously [54]. The identity of the protein products was confirmed by SDS-PAGE, followed by mass spectrometry.

For expression in mammalian tissue culture cells, BFSP1 constructs were subcloned into pEGFP-N3 (http://www.clontech.com/UK/Support/Applications/Using_Fluorescent_Proteins/EGFP_Vectors, accessed on 14 May 2023; Mountain View, CA, USA) with the eGFP added to the C-terminal end of the expressed BFSP1 construct as detailed previously [18].

### 2.3. Extraction of Lipids from Bovine Lens Cell Membranes

Bovine lens membranes were prepared as described [55]. The aqua-dissection method allowed membrane fractions to be prepared from the cortex and nucleus of the bovine lens resulting in four fractions corresponding to 18–11 mm, 11–8 mm, 8–6 mm and lastly 6–0 mm. Lens lipids were then extracted from 40 to 50 mg wet weight of each of these isolated bovine lens cell membranes using a Bligh and Dyer method [56]. Bovine lens cell membranes (10 mg/mL) per sample were centrifuged, and pellets were resuspended in 1 mL of 10 mM NaPO_4_, 100 mM NaCl, 5 mM EDTA, and pH 7.4 buffer. Samples were incubated for 30 min with 2 mL of dichloromethane: methanol (vol ratio 1:2). After incubation, 0.67 mL dichloromethane was added and 1.2 mL of 0.9% (*w*/*v*) KCl. Samples were centrifuged at 1000× *g* for 1 min. The lower phase was transferred to a new centrifuge tube and evaporated under nitrogen. Samples were stored at −20 °C.

### 2.4. Purification of Lens Membrane Lipids for SPR

Samples of the extracted bovine lens membrane lipids were warmed for 5 min at room temperature prior to incubating for a further 5 min at 37 °C. They were then resuspended in 10 mM NaPO_4_, 150 mM NaCl, pH 7.8, to a final concentration of 6.8 mg/mL. The resuspended lipid samples were resuspended mechanically both by alternating vortex and pipet for 5 min and then incubated for a further 10 min at 37 °C. The lipid samples were then heated to 90 °C and then allowed to cool on the bench to room temperature prior to 20 min sonication in a water bath and a final 5 min incubation at 37 °C. Immediately prior to use and the lipid preparation was mechanically agitated for 5 min. A 1:1 mixture of the lipids extracted from the lens membranes corresponding to the 11–8 mm and 8–6 mm regions of the lens was used to coat the L1 chip as per the manufacturer’s instructions (GE Healthcare).

### 2.5. Surface Plasmon Resonance to Measure Lipid-Binding of BFSP1 Constructs

Surface Plasmon Resonance (SPR) was used to assess the membrane binding properties of BFSP1 and its various domains. SPR experiments were conducted on a Biacore X100 (GE Healthcare, Uppsala, Sweden) using an L1 chip (GE Healthcare). Biacore X100 Control Software (Version: 2.0.1) with the Plus upgrade software, using the Kinetics/Affinity analysis workflow strategy according to the manufacturer’s instructions with 4 analysis cycles. The molecular weights used for the different BFSP1 constructs were 434–665 (44 kDa); 434–548 (26 kDa) and 494–550 (5.9 kDa). Molecular weights for human BFSP1 (74.5 kDa), bovine BFSP1 (83.1 kDa) and human CRYAB (20.2 kDa) were calculated using the online Expasy web resource (https://web.expasy.org/compute_pi/, accessed on 11 May 2023). All except the bovine BFSP1 were recombinantly produced proteins. CRYAB is a lens protein with membrane binding potential [57,58] and was used as a positive control. The L1 chip surface was cleaned by injecting 60 mM NaOH: Isopropanol (vol 3:2) and washed thoroughly with 10 mM NaPO_4_, 150 mM NaCl, pH 7.8. Liposomes (0.5 mg/mL) prepared from extracted lens membrane lipids were deposited on sensor chips at a low flow rate (10 μL/min) until a stable resonance unit (RU) level was obtained. Five different concentrations of full-length BFSP1 (human and bovine) and human BFSP1 fragments derived from the C-terminal domain (434–665, 434–548 and 494–550) in 10 mM NaPO_4_, 150 mM NaCl, pH 7.8 buffer were injected. The lipid surface was regenerated with 60 mM NaOH: isopropanol (vol ratio 3:2), providing a consistent baseline that was subsequently observed. All analyses were performed at 25 °C. Experiments determining the dissociation constant (K_D_) values were performed in triplicate.

### 2.6. Circular Dichroism Measurements

CD spectra were recorded from 280 to 180 nm with a Jasco-1500 spectropolarimeter (JASCO International Co. Ltd., Tokyo, Japan). Spectra were acquired at 20 °C using a 0.1 cm path length quartz cell, averaged over three scans at a scan speed of 20 nm/min, a bandwidth of 3.0 nm, 8 s digital integration time, and 0.5 nm resolution. Following baseline correction, the observed ellipticity, θ (mdeg), was converted to molar ellipticity, [Θ] (deg cm^2^/dmol), using the relationship [Θ] = θM/10(*lc*) where ‘*l*’ is the path length in centimetres, ‘*c*’ is concentration in g/L, and M is the average molecular weight (g/mol). All spectra were corrected with the appropriate blank. The peptides corresponding to residues 434–467 of BFSP1 were chemically synthesized ±N-terminal myristoyl group, then purified and characterised by Mass Spectrometry by the company Intavis Bioanalytical Instruments Aktiengesellshaft (Köln, Germany). Purified peptides were dissolved in water and trifluoroethanol (TFE; 30% (*v*/*v*)) to induce secondary structure formation [59], as previously suggested, to study the amphipathic helix in bee venom [60]. The α-helical content in 30% (*v*/*v*) TFE was determined using the online analysis resource (DiChroWeb: http://dichroweb.cryst.bbk.ac.uk/html/home.shtml, accessed on 4 May 2023; Version 1; London, UK [61]) using the algorithm CDSSTR [62] and reference set 7 of proteins [63].

### 2.7. Proteomic Analyses of Bovine Lens Membrane Preparations

Trypsin was added in a 1:50 weight/weight ratio to a bovine lens membrane sample. Typically, 50 µg of lens membrane protein was digested by overnight incubation at 37 °C. The digestion was terminated by adding trifluoroacetic acid (TFA). The final clean-up and fractionation of each digest were made using StageTips contained two SCX filter pieces as described [64]. Peptide analysis by LC-MS was performed on a SCIEX TripleTOF 6600 mass spectrometer linked to an Eksigent nanoLC 425 chromatography system via a 50-μm ESI electrode in a DuoSpray source (SCIEX). Sample injection and peptide separation used a trap and eluted method flowing at 5 µL/min. Peptides were loaded and washed on a YMC TriArt C18 guard column (1/32, 5 µm, 5 × 0.5 mm) followed by online chromatographic separation performed over 33 min on a YMC TriArt C18 column (1/32″, 12 nm, S-3 μm, 150 × 0.3 mm). A linear gradient of 5–35% (*v*/*v*) acetonitrile, 0.1% (*v*/*v*) formic acid over 20 min was then shifted to 80% (*v*/*v*) acetonitrile, 0.1% (*v*/*v*) formic acid over 2 min. The column was washed for 3 min before returning to 5% (*v*/*v*) acetonitrile, 0.1% (*v*/*v*) formic acid over to 2 min and re-equilibration for 6 min. Data-dependent LC-MS-MS acquisition was programmed for the time period 0.1–24 min. Once completed, the source voltage was lowered from 5500 to 0 V. Each MS cycle comprised a 250 ms precursor-ion scan from 350 to 1500 *m*/*z* followed by fragmentation of up to 10 selected ions for 100 ms each. This generated MS/MS spectra of 100 to 1500 *m*/*z* (cycle time 1.3 s). The switch criteria applied were +2 to +5 ions of intensity > 400 cps, with a rolling exclusion time of 12 s.

LC-MS/MS peak lists in (.mgf) format were generated from (.wiff) format data files using MSConvert in the ProteoWizard suite of proteomic software tools and peptide/protein identification used PEAKS X+ software (Bioinformatics Solutions Inc., Waterloo, ON, Canada). Default precursor and product ion tolerances for a TripleTOF spectrometer were specified, along with the following modifications: fixed—carbamidomethyl [C], variable—oxidation [M], acetylation [N-term], deamidation [NQ], oxidation [HW]. Search databases were a UniProt bovine reference proteome (March 2019, 22,285 entries) plus 194 known proteomic experiment contaminants, or a human reference proteome (February 2017, 20,106 entries) plus 161 contaminants. Peptide FDR was set at 1%, and a filter of 2 unique peptides was applied to identified proteins.

### 2.8. SDS PAGE and Immunoblotting

Protein samples were resuspended in SDS sample buffer (1 mM EDTA pH 7.8, 50 mM Tris-HCl pH 6.8 and 1% (*w*/*v*) SDS) and then mixed with 4× Laemmli sample buffer (containing 2.854 M β-mercaptoethanol) ([65] as modified in [18]). Protein samples were separated on either 10, 12 or 15% (*w*/*v*) polyacrylamide gels at 100 V depending on the relative electrophoretic mobility of the sample under investigation. Protein standards (PageRuler Plus, 10–250 kDa; ThermoFisher Scientific, Warrington, UK) were included. Coomassie blue staining (0.25% (*w*/*v*) Coomassie Brilliant Blue R250, Merckmillipore, Watford, UK) was used for protein detection, and once de-stained gels were imaged using either an Azure 300 or Thermofisher Scientific iBRIGHT imaging systems.

For immunoblotting, proteins separated by SDS-PAGE were transferred to nitrocellulose membrane for blotting using the semi-dry method, as detailed previously [18]. Briefly, after Ponceau dye staining to check the transfer, the membrane was blocked with 10% (*w*/*v*) milk powder and 6% (*w*/*v*) bovine serum albumin (Fraction V, Merck) in Tris-HCl Buffered Saline (TBS) at room temperature, for 2 h. Next, membranes were probed with a commercially available BFSP1 rabbit polyclonal antibody, the Atlas Antibody S38 (HPA042038-100UL, Sigma–Aldrich). Primary antibody binding was detected using EIA Grande Anti-Rabbit IgG Horseradish Peroxidase Conjugate (Cat. No. 172-1019) diluted in 3% (*w*/*v*) BSA in TBS. A signal was detected using the Azure 300 imaging system by the chemiluminescent method.

### 2.9. Eukaryotic Cell Culture, Incubation with Recombinant BFSP1 Fragments, Transient Transfections of BFSP1 Constructs and Live Cell Imaging

The mammary carcinoma cell line, MCF7, was grown in DMEM supplemented with glutamine, penicillin, streptomycin and 10% (*v*/*v*) foetal bovine serum. MCF7 cells were seeded on glass coverslips in six-well plates prior to transient transfection of the peGFP-N3 vectors containing various BFSP1 constructs, including the G434A mutation, to prevent myristoylation of the BFSP1 434–548 fragments. Purified plasmids were mixed with the GeneJuice^TM^ transfection reagent (Novagen; http://www.emdmillipore.com/life-science-research/novagen/, accessed on 14 May 2023, Watford, UK) according to the manufacturer’s guidelines. Cells were left overnight before live cell imaging by light microscopy using Zeiss LSM 710 confocal microscope in a chamber maintained with 5% (*v*/*v*) CO_2_ and at 37 °C.

To identify which intracellular membrane compartments of MCF7 cells could be labelled with eGFP tagged BFSP1 434–548, 434–460 and 494–550, cells were transiently transfected using Lipofectamine 2000 according to the manufacturer’s instructions (Thermofisher Scientific, UK). After overnight incubation, an hour before starting live cell imaging, the following vital dyes were added; Bodipy 493/505 (2 μL/mL), Hoescht 33342 (H33342) 2 μL/mL and Mitotracker 1/3000 either alone or in combination with each other according to manufacturer’s instructions (Thermofisher Scientific, UK). In addition, in some cases, transiently transfected cells were treated with 10 µM tamoxifen (MP Biochemicals) to induce plasma membrane blebbing [66].

## 3. Results

### 3.1. The C-Terminal Sequences of BFSP1 Contain an Amphipathic Helix That Spans the Myristoylation Motif and Predicted α-Helix

Human BFSP1 is a 665-residue protein. The N-terminal sequences contain an α-helical-rich region and C-terminal sequences in which there is a highly conserved caspase site (430–433) and myristoylation signal (434–439, Figure 1A and Appendix A).

Both the myristoylation and, therefore, by inference so too the caspase cleavage site, have been confirmed by mass spectrometry [20]. This is an example of post-translational myristoylation and was predicted by a recently developed search engine SVMyr for such events [23]. Using two different caspase prediction programmes (GrabCas [67] and CasCleave 2 [68]), three caspase sites were identified within residues 430–665 for human BFSP1 by the two programmes (Appendix A), but only two in the equivalent region of Bovine BFSP1 (Appendix A). We confirmed the presence of the D433 site but also identified a second caspase cleavage site with the recognition sequence SFVD (Bovine BFSP1 477–480; Appendix A). These data suggest that the other caspase sites identified within the C-terminal sequences of BFSP1 are potentially bone-fide cleavage sites in addition to the one adjacent to the myristoylation sequence. These need to be confirmed experimentally, as we have done here for the D480 site in bovine BFSP1, but collectively these data and the bioinformatic predictions suggest that both the caspase cleavage and myristoylation of the C-terminal fragments released from BFSP1 are important to its function and the regulation of AQP0 [57,69].

Interestingly, AlphaFold2 predicted an α-helix overlapping the myristoylation signal and spanning the region 434–452 (Figure 1B). The remaining C-terminal sequences lacked other secondary structures according to that prediction. The algorithm HELIQUEST is used to identify potential lipid-binding domains (LBDs) by combining a Discrimination Factor with the presence of α-helices (Figure 1C). That bioinformatic analysis strongly suggested the presence of an amphipathic helix spanning residues 434–452 (See Appendix A for full HELIQUEST analysis of sequences 434–665). Using the fIDPnn algorithm (Figure 1D), the 434–452 region, similar to the rest of the C-terminal sequences, is predicted to be intrinsically disordered, which does not entirely support the AlphaFold2 prediction for the 434–452 sequences. The high probability that a predicted α-helix would combine to produce an amphipathic helix was, however, reinforced by the MemBrain3 algorithm designed to identify such motifs (Figure 1E). The amphipathic helix prediction was consistent between human, rat and cow BFSP1, with a major peak between 434 and 454 and a shoulder extending to residues 466–468 between the three mammals (Figure 1E), which corresponds to the Heliquest discrimination factor remaining above the 0.68 threshold [45] until residue 464 (Figure 1A).

To confirm the presence of an α-helix and to monitor the influence of the myristoyl group on their structure, peptides covering residues 434–467 were chemically synthesised. CD spectroscopy revealed a disordered peptide with no α-helical content when dissolved in water (Figure 2A), but in the presence of 30% (*v*/*v*) TFE, the spectra for both the myristoylated and unmyristoylated peptides changed to include distinct minima (Figure 2B).

It was noticeable that the presence of the myristoyl group enhanced the spectra for the 434–467 peptide with the minima shifting from 204.5 to 206 and a distinct minimum now at 221 nm suggests α-helical structure (Figure 2B,C) as observed for other amphipathic helices [42,60] including ARF1 [39] and BAR-domain peptides [70]. However, such a secondary structure only becomes apparent when the CD spectra are measured in the presence of TFE and other reagents, such as SDS, that promote the formation of amphipathic helices [60].

### 3.2. The BFSP1 C-Terminal Sequences Distal to 434 Bind Lipid Membranes Independently of AQP0

For these studies, human BFSP1 and various fragments were produced recombinantly in *E.coli* (Figure 3). These included full-length BFSP1, 1–433, 434–665, 434–548 and finally, 494–550.

It is noticeable that the fragments from the BFSP1 C-terminal domain all migrated with slower-than-expected relative mobility according to their calculated molecular weights. We interpreted this as the most likely dimerization of the 434–665, 434–548 and 494–550 fragments, given the expected effects of SDS on the amphipathic helix that spans residues 434–447 (Figure 1C,E) and the predicted protein-protein interactions in the region 519–551 (Figure 1D; protein binding indicator). The fragment 434–548 was selected because of an additional predicted caspase site at residues 546–549 in Human BFSP1 (Appendix A) and the 494–550 as a control where no amphipathic helices or LBDs were predicted by the HELIQUEST algorithm (Figure 1C). These fragments (Figure 4A–C), as well as full-length bovine and human BFSP1, were then used to measure by SPR their relative affinities for a lipid bilayer derived from lens membranes formed on the L1 chip. Examples of the sensorgram plots and corresponding equilibrium dissociation constant, K_D_ calculations are provided for the recombinant BFSP1 fragments 434–665 (Figure 4A), 434–548 (Figure 4B) and 494–550 (Figure 4C).

Table 1 summarises the calculated K_D_ for these fragments and for full-length bovine BFSP1 purified from bovine lenses and recombinant human BFSP1. From these data, the BFSP1 434–548 fragment had the greatest affinity as compared to both full-length and other C-terminal sequences of human BFSP1. In addition, removing the sequences 549–665 increased the affinity of the 434–548 fragment for the eye lens lipids compared to 434–665 (Figure 4A,B, respectively; Table 1).

The sequences 494–550 (Figure 4C, Table 1) were similar in affinity to that of recombinant human CRYAB (Appendix A), a lens protein also identified with membrane binding properties [57,58]. Interestingly, full-length human BFSP1 and native bovine BFSP1 had similar affinities to the 434–665 fragment (Table 1), suggesting that sequences within the 434–665 region could be responsible for the binding of both full-length BFSP1 to lens membrane lipids (Figure 4A,B; Table 1). In addition, their affinity was similar to that observed for both the myristoylated and unmyristoylated polypeptides spanning residues 434–467 in human BFSP1 (Figure 4D,E respectively compared to Figure 4F; Table 2).

These data suggest that myristoylation had little effect on the affinity for lens lipids (Table 2). However, it is noticeable that the recombinant fragment spanning residues 434–548 had the greatest affinity for bovine lens membrane lipids (Table 1) compared to the polypeptides spanning 434–467 (Table 2), suggesting that there could also be a contribution from the sequences 467–548 to membrane binding.

### 3.3. Transient Transfections of eGFP Labelled BFSP1 Constructs Confirms Membrane Association for Fragments Containing the Amphipathic Helix

The transient transfection of a human epithelial cell line MCF7 with eGFP-tagged BFSP1 C-terminal sequences was undertaken to evidence that these constructs (Figure 5A) could associate with membranes in living cells (Figure 5B).

Only those C-terminal sequences that contained the amphipathic helix and adjacent C-terminal sequences localised to the plasma membrane and internal membranes of transfected cells (Figure 5B). These were the fragments 434–548, 434–665 and 434–460. It is worth noting that for all three constructs, it was not only the plasma membranes, but also large cytoplasmic vesicular structures that were eGFP positive. The nuclear membranes, however, remained negative (Figure 5B), suggesting there was selectivity in the observed membrane binding. The BFSP1 C-terminal fragments 461–548 and 550–665 did not localise to any membrane compartments, but their eGFP signal remained in the cytoplasmic compartment (Figure 5B). Preventing myristoylation by transfecting the mutant G434A-548 did not prevent its membrane localisation.

To identify which intracellular compartments were targets for labelling with the BFSP1 sequence 434–548, the vital dyes, Bodipy, Mitotracker and Lysotracker were used to counter label MCF7 cells and then followed by live cell imaging (Figure 6, red channel). Hoechst 33342 was used to label the nucleus (Figure 6, blue channel).

Intracellular vesicles were Bodipy positive (Figure 6 control, red channel), and these were also eGFP positive when transiently transfected with 434–548, but not 494–550. Mitochondrial membranes were negative but lysosomal membranes were positive (Figure 6; green channel) when transfected with 434–548 (Figure 6; arrows, green channel). In some cases, tamoxifen treatment was used to induce plasma membrane blebbing [66], and those vesicular structures that appeared to be extruding from the plasma membrane were also positive for eGFP BFSP1 434–665 and eGFP BFSP1 434–548, but not eGFP BFSP1 494–550 (Figure 6; green channel). These transfection data support the conclusion that the BFSP1 434–548 sequences can direct the association of BFSP1 to membrane compartments in living cells.

## 4. Discussion

### 4.1. BFSP1 Contains an Amphipathic Helix at Residues 434–452

The aim of this study was to determine whether the BFSP1 C-terminal domain of BFSP1 had the capacity to bind lens membrane lipids independently of AQP0, an integral membrane protein shown to interact with BFSP1 by chemical cross-linking [31]. We generated data from CD spectroscopy (Figure 2), SPR (Figure 4, Table 1 and Table 2) and transient transfections of eGFP-tagged BFSP1 constructs (Figure 5 and Figure 6) that support this conclusion. The bioinformatic analyses (Figure 1) suggested that a reason for the independent association of the C-terminal sequences of BFSP1 with lens fibre cell membranes [16,20,24,26,34,35] is the presence of an amphipathic helix spanning residues 434–451. Furthermore, the CD spectroscopy data show that a chemically synthesized polypeptide spanning residues 434–467 adopted an intrinsically disordered structure in water (Figure 2) in line with the fIDPnn prediction (Figure 1D) and that the inclusion of TFE [59] induced an α-helical transition (Figure 2) supporting the Heliquest amphipathic helix prediction (Figure 1C, Appendix A). SDS can also provide the hydrophobic environment needed to induce the transition from unstructured to an α-helical domain for amphipathic helices, and we suggest this contributes to the unexpectedly slower relative mobility observed by SDS PAGE of the recombinantly expressed fragments (Figure 3). This important detail was not considered in earlier studies concerning the proteolytic processing of BFSP1 [24,34] and points to the oligomerisation and membrane-directed scaffolding potential for these BFSP1 C-terminal fragments containing the amphipathic helix and the adjacent intrinsically disordered sequences.

The presence of an amphipathic helix is essential for membrane association and membrane remodelling activities. For example, the small GTPases, Atg proteins, ENTH/ANTH- containing proteins and the BAR-domain all require amphipathic helices for their function [71]. Rhodopsin recovery depends upon recoverin binding to an amphipathic helix in its N-terminus [72]. In a broader neurological context, synapse evolution and function owe much to the protein complexin and its amphipathic helix [73]. The amphipathic helix is key to the membrane association and targeting of such proteins as ARF1 [40,42] and other ARF family members [41]. In the case of Arfrp1 and Arl14, their amphipathic helices are sufficient to target these proteins to their appropriate membrane compartment [41]. Interestingly, naturally occurring mutations have been reported in amphipathic helix-containing membrane fission-inducing proteins and are the basis of human diseases [71] such as centronuclear myopathy (amphiphysin 2), Charcot-Marie-Tooth disease (GDAP1), Parkinson’s disease (α-synuclein). It is perhaps no coincidence that when the C-terminal domain of BFSP1 is deleted, cataract develops [74], and there are many sequence variants listed for BFSP1 434–548 (https://www.uniprot.org/uniprotkb/Q12934/entry#disease_variants, accessed on 10 May 2023) some such as E468Q listed as likely to be disease-causing, as well as 14 other variants within the amphipathic helix itself that deserve further investigation. For example, Q436R, Y445C and R446S all change the hydrophobic moment of the amphipathic helix in BFSP1, changing the Discrimination Factor scores (Table 3), indicating such sequence variants could have both structural and functional consequences worthy of further investigation.

### 4.2. Potential Contribution of the Amphipathic Helix and Myristoylation to BFSP1 Structure and Function in the Lens

On binding to a membrane, the amphipathic helix will usually orient itself parallel to the membrane plane. Its hydrophobic side will insert into the bilayer, while the hydrophilic face will interact with the headgroups of the lipids in the membrane [71]. The lipid composition of the lens plasma membranes depends upon the differentiation status of the fibre cells, which is related to whether they are present in the lens nucleus or the lens cortex and the age of the lens [75]. Our studies used lipids extracted from the nucleus of bovine lenses and, therefore, will be enriched in cholesterol and sphingolipid. Clearly, it will be important to investigate how the lipid composition affects BFSP1 binding, but our transient transfection studies evidence that the amphipathic helix in BFSP1 is sufficient to direct membrane association of eGFP in MCF7 cells (Figure 5 and Figure 6). Our data also suggest that other sequences within the 434–548 region likely influence the infinity (cf. Table 1 and Table 2). Our data show that there is membrane selectivity as neither the nuclear nor the mitochondrial membranes were labelled (Figure 6). Intracellular vesicles and the lysosomal membranes were, however, labelled (Figure 6). Both are critical to lens differentiation in organelle removal [76,77] and for aquaporin trafficking by a lysosomal-associated unconventional protein secretion mechanism [78]. Then the process of lens fibre cell compaction [79] and the fenestration of lens fibre cells leading to the formation of a stratified syncytium [80] will require an amphipathic helix containing proteins to be involved. Epitope mapping of bovine BFSP1 showed that the very C-terminal region becomes concentrated on the cell membranes in the bovine lens nucleus [24], which in the context of the data presented here, provides a different perspective on how beaded filaments and aquaporins affect the optical properties of the lens [18,33,81].

BFSP1 is the first intermediate filament protein identified to possess an amphipathic helix and the first to be identified that undergoes post-translational myristoylation. Our data are consistent with the view that myristoylation is not necessary for the membrane association of the 434–548 fragment (Figure 5B), but our previous study has demonstrated that myristoylation of this BFSP1 fragment is important to its regulation of AQP0 [18]. The presence of an amphipathic helix provides yet another mechanism to mediate the membrane association of an intermediate filament protein. Periplakin links vimentin, and the BFSP1 assembly partner BFSP2 to lens membranes [82] and is part of the EPPD complex [83], as well as plakoglobin [84] that links partners to the plasma membrane via the cadherins [83]. Whilst it is recognised that intermediate filaments impart physical properties to the membranes they support [85], there is also a growing appreciation that non-filamentous forms of cytoplasmic intermediate filament proteins, such as vimentin, have functions in the extracellular space [86,87]. The release of a C-terminal fragment of BFSP1 that contains an amphipathic helix capable of association with intracellular vesicles and with secreted membrane vesicles evidences a new cellular role for BFSP1.

## 5. Patents

The work described here forms the basis for the following patents that are pending (PCT/HU2021/050057 WO 2022/090752).

## Figures and Tables

**Figure 1 cells-12-01580-f001:**
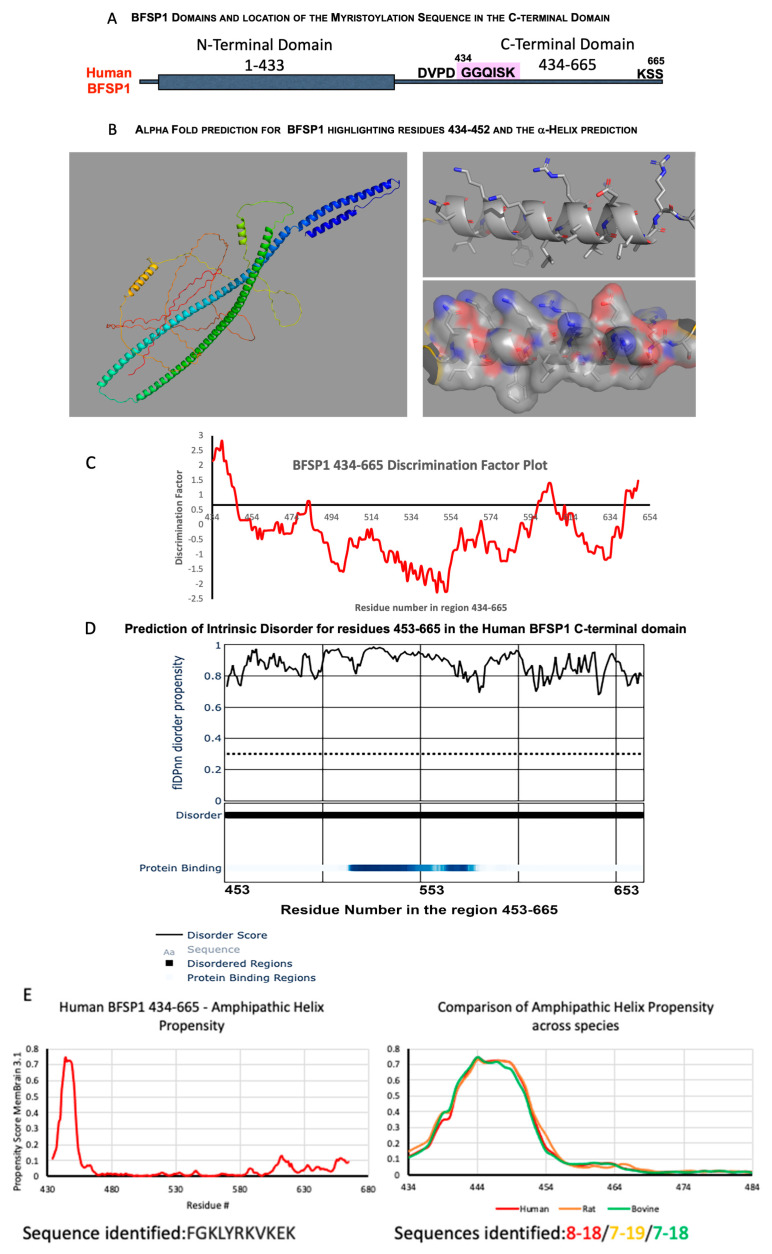
Bioinformatic analyses of BFSP1 and its C-terminal sequences 434–665. Human BFSP1 is a 665-residue protein, with a large α-helical rod domain (grey rectangle) and a C-terminal domain containing a cryptic myristoylation site (GGQISK) that is released by caspase cleavage at site 433 (**A**). AlphaFold2 predicted with high confidence an α-helix for residues 434–452, whilst the remaining C-terminal sequences had low/very low secondary structure prediction (**B**). HELIQUEST was used to calculate the discrimination factor (DF) over the region 434–665, with the X-axis intersect set to a DF of 0.68 as the threshold for identifying LBDs. The strongest score was received for the sequence 434–453 (**C**), but others were identified starting at residues 597 and 641 in the absence of predicted α-helix. The C-terminal BFSP1 sequences (454–665) were predicted to have a considerable intrinsic disorder as predicted using the algorithm fIDPnn (**D**). A minimum threshold score of 0.3 is indicated (---), but most of the BFSP1 453–665 sequences scored ≥ 0.8, indicative of significant disorder potential (**D**). A region potentially involved in protein-protein interactions was identified between 520 and 550 (dark blue bar; **D**). The 434–665 sequences were assessed for their propensity to form amphipathic helices, and one region was identified within the BFSP1 region 434–455 (**E**). The same region was identified in rat and bovine sequences (**E**), in line with the observation that the caspase recognition site and the myristoylation sequences are highly conserved across species [18].

**Figure 2 cells-12-01580-f002:**
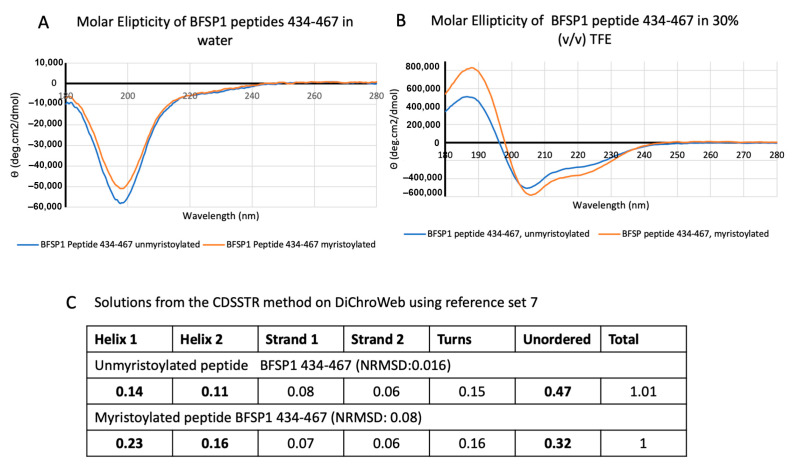
CD spectrum for myristoylated and unmyristoylated peptides spanning residues 434–467 in BFSP1. The CD spectra for chemically synthesized peptides spanning the residues 434–467 of human BFSP1 ± a myristoyl group were measured in water (**A**) and in (**B**) 30% (*v*/*v*) TFE. Peptides (0.05 mg/mL) were measured in a Jasco Quartz cell (0556) with a 0.1cm pathlength. Spectra shown are buffer blanked. Note the shift in the minima (206/220 nm) for the myristoylated peptide, which is reflected in the increased α-helical content as indicated from the CDSSTR outputs (**C**).

**Figure 3 cells-12-01580-f003:**
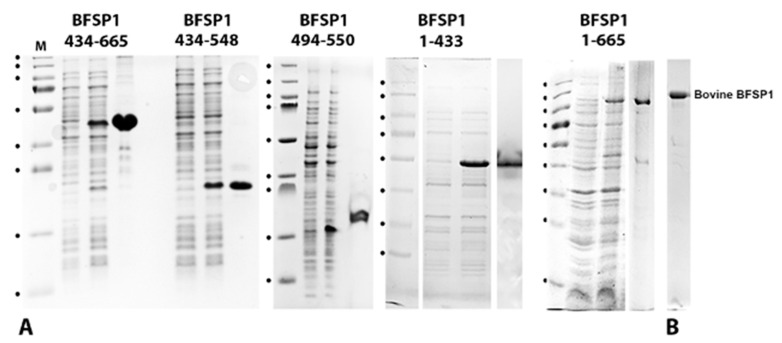
Recombinant expression and purification of the BFSP1 protein domains used in these studies. (**A**). BFSP1 and the domains 434–665, 434–548 and 494–550 were cloned into pET vectors, and their expression was induced by the addition of IPTG. In each of the panels, a sample of the uninduced, followed by a sample after a 3 h induction with IPTG and the subsequent affinity purified proteins are shown. The same markers (M) were used, although the acrylamide concentration changed; 10% (*w*/*v*) for BFSP1 1–665; 12% (*w*/*v*) for BFSP1 434–665 and 434–548; and lastly, 15% (*w*/*v*) for BFSP1 494–550. Markers correspond to 250, 130, 100, 70, 55, 35, 25, 15, and 10 kDa, respectively. (**B**). Bovine BFSP1 purified from bovine lenses.

**Figure 4 cells-12-01580-f004:**
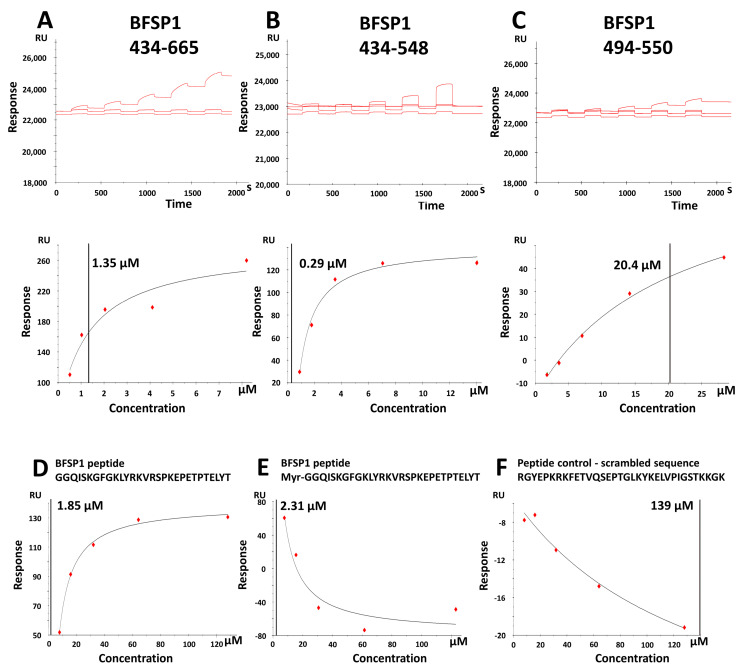
SPR measurement of the KD for BFSP1 C-terminal domain sequences. The sensorgram and calculated affinity (Equilibrium dissociation constant; K_D_) of the BFSP1 fragments 434–548 (**A**), 434–548 (**B**) and 494–550 (**C**) for L1 biosensor chip loaded with bovine lens membrane lipids over a five-point concentration range. The plots of RU against concentration for BFSP1 434–665 (**A**), 434–548 (**B**) and 494–550 (**C**). The fragments were produced recombinantly in *E.coli,* and the final protein preparations after SEC purification used for the SPR experiments are shown in the insets of each panel. The chemically synthesized peptides of BFSP1 434–467 ± myristate (**D**,**E**) were compared to a scrambled peptide control (**F**). In all the plots (**A**–**F**), the K_D_ value is included in the plot and can be cross-referenced to Table 1 and Table 2.

**Figure 5 cells-12-01580-f005:**
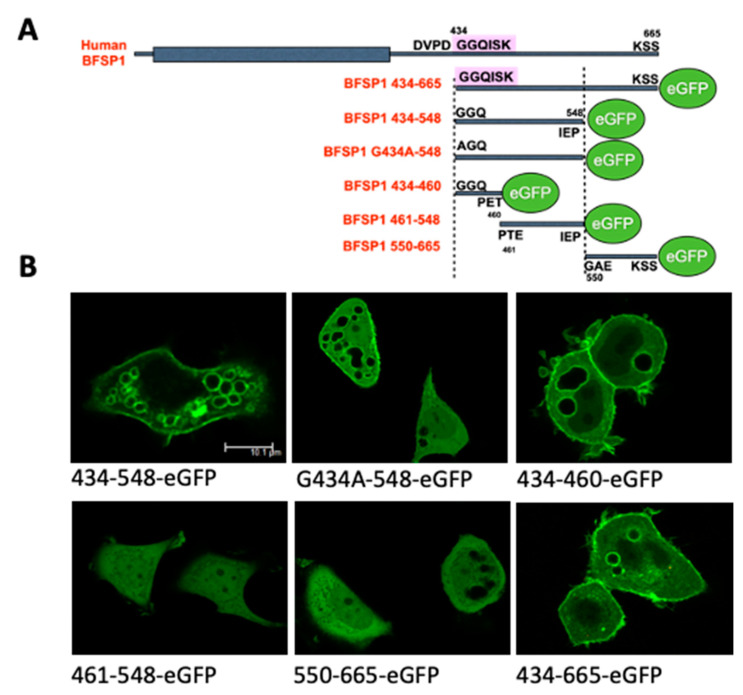
Transient transfection of C-terminal domain fragments from BFSP1 confirms 434–460 as a bone fide LBD. A series of eGFP-tagged BFSP1 constructs derived from the C-terminal domain designed to confirm the membrane binding of the amphipathic helix associated with the region 434–452 were subcloned into the peGFP-N3 vector (**A**). Transient transfection of these different constructs into MCF7 cells produced prominent plasma membrane staining for some (**B**; 434–665; G434–P548; A434–P548), but not all constructs (**B**; 461–548, 550–548). Furthermore, preventing myristoylation by replacing the N-terminal glycine residue with alanine (G434A) did not prevent the plasma membrane localization of BFSP1 434–548, suggesting that myristoylation was not necessary for its plasma membrane, or internal membrane localisation. Scale bar = 10 µm (**B**).

**Figure 6 cells-12-01580-f006:**
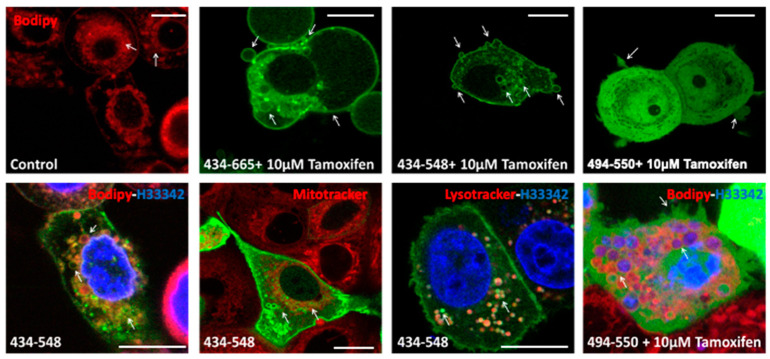
Localisation of BFSP1 434–548 with the lysosomal compartment. MCF7 cells were transiently transfected with BFSP1 434–548 and 434–665 tagged with eGFP and monitored by lice cell imaging. Transfected cells were labelled with the vital dyes, Lysotracker, Mitotracker, Bodipy 494/503 (Bodipy) and Hoechst 33324 *H33324) to follow the distribution of the fragments 434–548, 494–550 and 434–665 in transiently transfected MCF7 cells. Bodipy labelled internal membranes (arrows; Control), and in BFSP1 434–548 transfected cells, these were also co-labelled with eGFP (434–548; arrows). Note the absence (arrows; 434–548) of co-staining of mitochondria (Mitotracker; green channel) and BFSP1 434–548 (green channel), but that the lysosomal marker, Lysotracker (red channel), overlapped (arrows; 434–548) with BFSP1 434–548 signal (green channel). Tamoxifen (10 µM) induced membrane blebbing resulted in 434–548 and 434–665 labelled blebs, but this was not seen in cells transfected with BFSP1 494–554. Scale bars = 10 µm.

**Table 1 cells-12-01580-t001:** SPR Measurement of KD for BFSP1 protein and C-terminal domain regions binding to Bovine Lens Fibre Cell Membrane Lipids.

Bovine BFSP1 (Native)	Human BFSP1 (Recomb.)	Human BFSP1 434–665	Human BFSP1 434–548	Human BFSP1 494–550	
6.02 × 10^−6^	1.32 × 10^−6^	3.06 × 10^−6^	1.60 × 10^−7^	1.05 × 10^−5^	
6.75 × 10^−6^	3.43 × 10^−6^	1.34 × 10^−6^	2.88 × 10^−7^	2.04 × 10^−5^	
7.52 × 10^−6^	3.94 × 10^−6^	2.19 × 10^−6^	2.24 × 10^−7^	1.53 × 10^−5^	
6.76 × 10^−6^	2.90 × 10^−6^	2.20 × 10^−6^	2.24 × 10^−7^	1.54 × 10^−5^	Mean
4.34 × 10^−7^	8.02 × 10^−7^	4.94 × 10^−7^	3.70 × 10^−8^	2.86 × 10^−6^	SEM

**Table 2 cells-12-01580-t002:** Effect of myristoylation on the lipid-binding of a polypeptide BFSP1 434–467.

Unmyristoylated BFSP1 434–467	Myristoylated BFSP1 434–467	Scrambled Polypeptide Control	
1.85 × 10^−6^	2.19 × 10^−6^	1.39 × 10^−4^	
2.42 × 10^−6^	2.31 × 10^−6^	1.67 × 10^−4^	
2.13 × 10^−6^	**2.25 × 10^−6^**	**1.53 × 10^−4^**	Mean
2.83 × 10^−7^	5.85 × 10^−8^	1.37 × 10^−5^	SEM

**Table 3 cells-12-01580-t003:** Change in the Heliquest Discrimination Factor as a result of reported sequence variants in the 434–464 sequence of BFSP1, which includes the amphipathic helix (grey highlight). In this table, three sequence variants (Q436R, Y445C and R446S) reported for human BFSP1 (https://www.uniprot.org/uniprotkb/Q12934/entry#disease_variants, accessed on 10 May 2023) are analysed using the Heliquest algorithm (https://heliquest.ipmc.cnrs.fr, accessed on 10 May 2023). The values for the Hydrophic moment (HM) for z (net charge) and then used to calculate the Discrimination Factor (DF) for the wild type and three variants as indicated. Those sequences where a hydrophobic face is predicted are indicated (grey highlight), and DFs that are either higher or lower than the wild type of sequence are indicated (bold).

BFSP1 434–454 (Wild Type)	Wild Type			Q436R			Y445C			R446S		
Peptide Sequence Scan	HM	z	DF	HM	z	DF	HM	z	DF	DM	z	DF
GGQISKGFGKLYRKVKEK	0.53932	5	2.15911808	0.56706	6	**2.51530464**	0.55949	5	**2.17815856**	0.49211	4	**1.78455184**
GQISKGFGKLYRKVKEKV	0.59755	5	2.2140872	0.62717	6	**2.57204848**	0.61913	5	**2.23445872**	0.55201	4	**1.84109744**
RISKGFGKLYRKVKEKVR	0.63442	6	2.57889248	0.66175	7	**2.934692**	0.65431	6	**2.59766864**	0.58691	5	**2.20404304**
ISKGFGKLYRKVKEKVRS	0.62845	6	2.5732568	0.62845	6	2.5732568	0.64803	6	**2.59174032**	0.58061	5	**2.19809584**
SKGFGKLYRKVKEKVRSP	0.57526	6	2.52304544	0.57526	6	2.52304544	0.59602	6	**2.54264288**	0.52872	5	**2.14911168**
KGFGKLYRKVKEKVRSPK	0.56039	7	2.83900816	0.56039	7	2.83900816	0.57893	7	**2.85650992**	0.51154	6	**2.46289376**
GFGKLYRKVKEKVRSPKE	0.54099	5	2.16069456	0.54099	5	2.16069456	0.55961	5	**2.17827184**	0.49222	4	**1.78465568**
FGKLYRKVKEKVRSPKEP	0.54667	5	2.16605648	0.54667	5	2.16605648	0.56336	5	**2.18181184**	0.49629	4	**1.78849776**
GKLYRKVKEKVRSPKEPE	0.4226	4	1.7189344	0.4226	4	1.7189344	0.43641	4	**1.73197104**	0.37039	3	**1.33964816**
KLYRKVKEKVRSPKEPET	0.41416	4	1.71096704	0.41416	4	1.71096704	0.42877	4	**1.72475888**	0.36327	3	**1.33292688**
LYRKVKEKVRSPKEPETP	0.35341	3	1.32361904	0.35341	3	1.32361904	0.36259	3	**1.33228496**	0.29968	2	**0.94289792**
YRKVKEKVRSPKEPETPT	0.28225	3	1.256444	0.28225	3	1.256444	0.29517	3	**1.26864048**	0.22951	2	**0.87665744**
RKVKEKVRSPKEPETPTE	0.26427	2	0.90947088	0.26427	2	0.90947088	0.26427	2	0.90947088	0.21098	1	**0.52916512**

## Data Availability

The data presented in this study are available on request from the corresponding author. The data are not publicly available due to privacy reasons.

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
