# Peer review of "Independent Membrane Binding Properties of the Caspase Generated Fragments of the Beaded Filament Structural Protein 1 (BFSP1) Involves an Amphipathic Helix"

_cells, 2023, doi:10.3390/cells12121580_

Round 1

Reviewer 1 Report

This study reports the membrane binding nature of the C-terminal fragment in BFSP1 protein involving helix structure. The experimental design using bioinformatics for predicting and CD for reporting helicity, SPR experiments to indicate binding with lens lipids, and transfecting epithelial cells to show the binding of protein fragments in the membrane is convincing. The manuscript provides enough evidence to support the findings and contains sufficient references (75). Thus, I have only minor comments.

1)      While extracting the bovine lens lipid, the whole lens was used. Cholesterol content in the cortex and nucleus membranes varies significantly. Is there any correlation of cholesterol in binding the BSFP1 fragment of interest with the membrane? Also, how many bovine lenses were used to extract lipids?

2)      In the CD spectroscopy experiment, peptides were dissolved in TFE to induce secondary structure formation, while sodium phosphate buffer was used for SPR experiments. Does it signify that the binding of BSFP1 fragments binds without forming a helix structure?

I also noticed a different title in the actual print and the one shown online.

Author Response

This study reports the membrane binding nature of the C-terminal fragment in BFSP1 protein involving helix structure. The experimental design using bioinformatics for predicting and CD for reporting helicity, SPR experiments to indicate binding with lens lipids, and transfecting epithelial cells to show the binding of protein fragments in the membrane is convincing. The manuscript provides enough evidence to support the findings and contains sufficient references (75). Thus, I have only minor comments.

Thank you for this very supportive review.

1)      While extracting the bovine lens lipid, the whole lens was used. Cholesterol content in the cortex and nucleus membranes varies significantly. Is there any correlation of cholesterol in binding the BSFP1 fragment of interest with the membrane? Also, how many bovine lenses were used to extract lipids?

We have added extra detail to the materials and methods (lines 150-154; 169-172). We thank the referee for noting that the lipid composition changes across the lens and we agree that this is important. This is why there is a second study currently underway to investigate BFSP1 binding to individual lipids. At this stage we are therefore unable to say whether cholesterol influences the binding. For completeness, 20-24 lenses from animals approx 6 months old were used to prepare the membranes prior to lipid extraction.

2)      In the CD spectroscopy experiment, peptides were dissolved in TFE to induce secondary structure formation, while sodium phosphate buffer was used for SPR experiments. Does it signify that the binding of BSFP1 fragments binds without forming a helix structure.

TFE was added to 30% (v/v). This is a standard approach to encourage the amphipathic helix to form and be measured by CD spectroscopy. This resembles the lipid environment and therefore TFE is not needed when adding the proteins and their fragments to the L1 chip itself. In other studies of amphipathic helices,  SDS has been used.  So the two methods employed are complimentary and not contradictory. The BFSP1 434-548 fragment binds to the membrane via an amphipathic helix. Removing the amphipathic helix negates the membrane binding.

I also noticed a different title in the actual print and the one shown online.

My apologies - The old dog new tricks scenario. I attempted to make the title change online but clearly did not do this successfully. Please accept my apologies - but we note the attention to detail by this referee.

Reviewer 2 Report

The manuscript by Miguel Jarrinand co-workers explores mechanisms of a unique protein fragment derived from the lens-specific BSFP1 protein and its interactions with fiber cell membrane independent of aquaporin 0/MIP channel protein established earlier. The experiments probe in depth the molecular processes, including all necessary controls. The data are shown in 6 Figures and 3 Tables and three supplemental figures. All experimental procedures are described with sufficient details and background references. It would be nice to include multiple sequence alignment of BFSP1 proteins from different species as an additional supplementary figure, including annotated domains discussed in the manuscript. Taken together, this manuscript addresses an important gap in our knowledge of lens structural biology and is very appropriate for publication in the special issue of Cells “New Advances in Lens Biology and Pathobiology” following some minor editing/inclusion of additional data. 

Minor comments:

1)    Title and Abstract: Include BFSP1 in the title and define aquaporin 0/MIP in the Abstract.

2)    Abstract (line 27): …of non-lens MCF7 …

3)    Introduction: Additional information on lens-specific BFSP1 and BFSP2 proteins and their roles in lens fiber cell development and cataractogenesis should be included. There is some information in Discussion but earlier broader background is always helpful.

4)    Some subtitles end with (.) while others not.

5)    Materials and Methods (line 136): produced recombinantly(where?) as described …

6)    Materials and Methods (line 207): correct typos (upper case letters).

7)    Materials and Methods (line 254): correct typo Cell (upper case letter). 

8)    Results: Figures 1 and 2 can be shown in a single page without being split over two pages.

9)    Table 1: Show within a single page (see above).

Author Response

The manuscript by Miguel Jarrin and co-workers explores mechanisms of a unique protein fragment derived from the lens-specific BSFP1 protein and its interactions with fiber cell membrane independent of aquaporin 0/MIP channel protein established earlier. The experiments probe in depth the molecular processes, including all necessary controls. The data are shown in 6 Figures and 3 Tables and three supplemental figures. All experimental procedures are described with sufficient details and background references. It would be nice to include multiple sequence alignment of BFSP1 proteins from different species as an additional supplementary figure, including annotated domains discussed in the manuscript. Taken together, this manuscript addresses an important gap in our knowledge of lens structural biology and is very appropriate for publication in the special issue of Cells “New Advances in Lens Biology and Pathobiology” following some minor editing/inclusion of additional data. 

We thank the referee for their positive comments and support t=for the submission.

Minor comments:

1)    Title and Abstract: Include BFSP1 in the title and define aquaporin 0/MIP in the Abstract.

Thank you - this we have done. 

2)    Abstract (line 27): …of non-lens MCF7 …

This change has been made

3)    Introduction: Additional information on lens-specific BFSP1 and BFSP2 proteins and their roles in lens fiber cell development and cataractogenesis should be included. There is some information in Discussion but earlier broader background is always helpful.

We thank the referee for this suggestion. We have added this requested background information (lines 45-49) with citations to help the interested reader access the published literature.

4)    Some subtitles end with (.) while others not.

Thank you - 4 instances corrected

5)    Materials and Methods (line 136): produced recombinantly(where?) as described …

Corrected - "E.coli" has been added

6)    Materials and Methods (line 207): correct typos (upper case letters). 

These have now been written in full and the abbreviation included.

7)    Materials and Methods (line 254): correct typo Cell (upper case letter).

Corrected - thank you 

8)    Results: Figures 1 and 2 can be shown in a single page without being split over two pages.

The Instructions request that the figures are inserted into the manuscript at the first time of mention. This creates such issues, but we will attend to this at the proof stage because we were not completely satisfied with the forced formatting.

9)    Table 1: Show within a single page (see above).

On the version I have downloaded this is now the case.